# OBJECT-AWARE CONDITIONAL ALIGNMENT FOR CROSS-DOMAIN COUNTING

## ABSTRACT

Object counting is an important task in computer vision with many real-world applications. In practical settings, factors such as lighting conditions and object density can vary dramatically, leading to distribution shifts then causing inaccurate counting. We found that existing domain adaptation (DA) methods cannot be directly applied to the counting task, as they usually assume changes across different domains are task-irrelevant and focus on utilizing domain-invariant features for prediction. However, in object counting tasks, changes in object density which could happen across domains are task-relevant and cannot be ignored. Therefore, applying existing DA methods to the counting task can ignore the information about density changes, resulting in unreliable counting. To address this limitation, we propose the Binary Alignment Network (BiAN). Unlike traditional DA methods that align distributions of entire image representations, BiAN segments objects of interest and aligns the distributions of the object-specific features across domains. This targeted alignment allows us to disregard irrelevant features, such as lighting conditions while preserving essential information about changes in object density. We theoretically demonstrate that BiAN achieves superior adaptability in counting tasks by introducing conditional alignment—aligning features conditioned on the presence of objects. Extensive experiments on two distinct counting tasks and eight dataset combinations show that BiAN outperforms state-of-the-art methods.

## 1 INTRODUCTION

Object counting is an important task in computer vision with a wide range of real-world applications, including crowd monitoring, traffic analysis, and biomedical imaging. Accurate counting of objects within images or video frames is crucial for decision-making processes in various industries and research domains (He et al., 2021). However, in practical settings, factors such as lighting conditions, object density, and background complexity can vary dramatically across different environments. These variations lead to distribution shifts between training data (source domain) and deployment scenarios (target domain), posing significant challenges for object counting models (Wang et al., 2019b).

To address distribution shifts, domain adaptation (DA) methods have been widely explored in machine learning. These methods aim to improve the generalization capabilities of models by aligning the feature distributions between source and target domains (Singhal et al., 2023). In tasks like image classification and semantic segmentation, DA methods generally assume that domain shifts are task-irrelevant, meaning the shifts do not affect the core features necessary for accurate predictions (Kong et al., 2022; Xie et al., 2023b; Bousmalis et al., 2016). By focusing on learning domain-invariant features, these methods strive to maintain performance across different domains.

However, this assumption does not hold in the context of object counting. Changes in object density across domains are inherently task-relevant, as the primary goal is to accurately estimate the number of objects

Figure 1: Comparison between existing domain adaptation (DA) methods and our approach. It shows that the general DA methods treat task-relevant factors as features that need to be directly aligned. The aligned distribution of density leads to consistent density estimation across domains. However, the consistent density does not match the real density in the samples. In our method, we only align the distributions of features belonging to objects of interest, so that the inter-object information can be preserved.

present (Li et al., 2019; Han et al., 2023). As shown in Figure 1, traditional DA methods that ignore these density variations may inadvertently discard crucial information, leading to unreliable counting performance on the target domain. The misalignment arises because these methods treat all domain shifts uniformly, failing to distinguish between task-relevant and task-irrelevant variations. The existing domain adaptive counting methods like CODA notice the issue of dynamic density (Li et al., 2019). However, they still consider the density feature as domain invariant and then struggle with aligning its distribution, which is in conflict with the assumption.

To tickle this limitation, we propose the Binary Alignment Network (BiAN), a novel approach designed specifically for domain adaptation in object counting tasks. Figure 1 shows the sketch of our proposed method. Instead of aligning the distributions of entire image representations, BiAN focuses on segmenting objects of interest and aligning the distributions of their features across domains. By isolating the features within the objects, our method effectively filters out task-irrelevant variations such as lighting conditions and background noise, while preserving essential information about object density changes. This targeted alignment ensures that the model remains sensitive to variations that directly impact the counting task.

We further provide a theoretical framework demonstrating that BiAN achieves superior adaptability by introducing conditional alignment—aligning features conditioned on the presence of objects. This approach allows the model to account for density variations between domains more effectively than traditional methods. By conditioning on object presence, BiAN maintains sensitivity to the number and arrangement of objects, which are critical factors in accurate counting.

Our main contributions are summarized as follows:

- We highlight the shortcomings of existing domain adaptation techniques when applied to object counting tasks. Specifically, the existing DA methods contempt the dynamic density across scenes as the task-relevant factor, violating the DA assumption. The aligned density distribution causes source-consistency density estimation, leading to decay in counting performance.

- We introduce BiAN, which segments objects of interest and aligns their features across domains conditionally. Within BiAN, the conditional alignment is proposed for aligning the distribution of object feature and background and preserving inter-object contextual information. The additional consistency mechanism further guarantees the crucial density information by constraining the conditional aligned distribution consistent with the distribution of the entire sample.

- We provide a theoretical demonstration of how conditional alignment enhances domain adaptability in counting tasks. The analysis demonstrates that aligning the distribution of specific conditions is beneficial to overall alignment performance. It also emphasizes the importance of maintaining consistency between condition-level distribution and sample-level distribution.

- We conduct comprehensive experiments on multiple counting scenarios with different density variations. The results show that BiAN significantly outperforms existing methods in terms of counting accuracy.

In the following sections, we review related work, detail the methodology of BiAN, present our theoretical findings, and discuss the experimental results that demonstrate the advantages of our approach.

## 2 RELATED WORK

### 2.1 OBJECT COUNTING

Object counting is a fundamental task in computer vision, with applications in various fields, such as crowd monitoring, cell counting, and traffic analysis (Loy et al., 2013). Traditional counting methods rely on supervised learning, which requires a large amount of annotated data (Jiang et al., 2021; Liu et al., 2021; Gao et al., 2021). Recent advances in deep learning have significantly improved the performance of counting models. For instance, Kernel-based Density Map Generation (KDMG) (Wan et al., 2022) employs a kernel-based density map to estimate the object count. SAU-Net (Guo et al., 2022) combines the advantages of SANet and U-Net to achieve high counting accuracy. STEERER (Han et al., 2023) cumulatively selects and inherits discriminative features to resolve scale variations. Despite the remarkable performance of these models, they are limited by the requirement of large amounts of annotated data when encountering domain variety. Therefore, GAN-based UDA counting methods have been proposed, such as Counting Object via scale-aware adversarial Density Adaptation (CODA) (Li et al., 2019), devised to address distinct object scale and density distributions. Additionally, SSIM Embedding Cycle GAN (SECycle) (Wang et al., 2019b) has emerged as a potent solution for counting in natural crowd scenes by synthesizing target-like images. To amplify the model's adaptability across intricate scenarios, the novel Latent Domain Generation (LDG) (Zhang et al., 2023) method has been introduced, generating the latent domain to learn the distribution from domains. The advanced research adopts the latest approaches in other fields, such as SaKnD (Xie et al., 2023a) which utilizing diffusion modules to enhance generalizability and CrowdGraph (Zhang et al., 2024a) which proposed an algorithm via pure graph neural network. To the best of our knowledge, there remains a research gap in discriminate migration for preserving task-relevant information across domains.

### 2.2 DOMAIN ADAPTATION

Domain adaptation (Cai et al., 2019; Courty et al., 2017; Deng et al., 2019; Tzeng et al., 2017; Wang et al., 2019a; Xu et al., 2019; Zhang et al., 2017; 2020; Mao et al., 2024; Stojanov et al., 2024; Wu et al., 2022; Eastwood et al., 2022; Tong et al., 2022; Kirchmeyer et al., 2022; Zhu et al., 2022; Xu et al., 2022; Roelofs et al., 2022; Kong et al., 2022; Jiang et al., 2022; Liu et al., 2022; Zhang et al., 2024b) has become a focal point in recent computer vision and machine learning research. Among the various approaches, invariant representation learning, introduced by (Ganin & Lempitsky, 2015), stands out as a direct and increasingly popular method. The goal of invariant representation learning is to identify domain-invariant features, that can reconstruct the original data for predicting label (Bousmalis et al., 2016). Historically, it was assumed

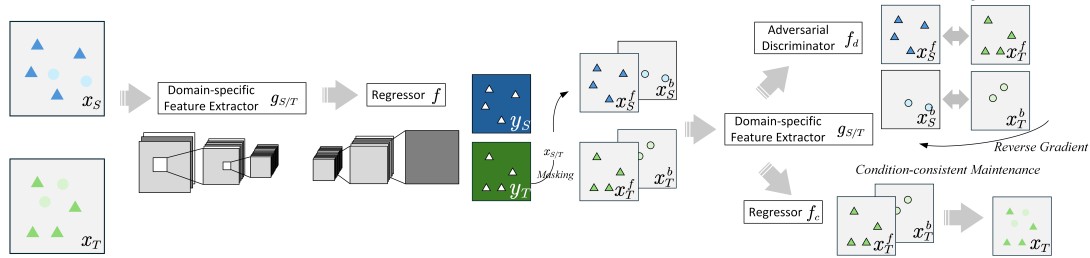

Figure 2: Overview of our proposed BiAN framework. $g_s$ and $g_t$ are domain-specific feature extractors for source and target domain. $f_d$ is the domain discriminator for aligning. $f$ is the regressor for generating target density map. $f_c$ is the regressor for generating conditional density map with shared weights with $f$.

that the distribution of labels remains consistent across different domains. Based on this assumption, cluster-based and kernel-based methods have been developed to approximate the joint label distribution (Long et al., 2018; Xie et al., 2018a; Shu et al., 2018). In general, it is hard to guarantee that domain-invariant features capture the discriminative information needed for label prediction in a setting of single source domain (Kong et al., 2022). Multi-source adaptation offers potential solutions (Xu et al., 2018; Peng et al., 2019; Park & Wan Lee, 2021; Li et al., 2021; Wang et al., 2020), where theoretical studies have demonstrated that latent variables can be identified from a sufficient number of source domains using independent component analysis. However, the existing methods are limited by the assumption that the domain shifts are task-irrelevant. In contrast, our proposed BiAN focuses on aligning the distribution of object-specific features across domains, which allows us to disregard irrelevant features while preserving essential information about object density changes.

## 3 METHODS

In this section, we formally propose the Binary Alignment Network (BiAN) for cross-domain counting tasks. We first describe the conditional alignment process in Figure 2 and discuss how we can mitigate the domain shift of object features while preserving crucial density information. We introduce conditional alignment and consistency mechanisms in Section 3.2 and Section 3.3. The training process of BiAN is shown as Appendix A.2. We also provide a theoretical analysis of conditional alignment in Section 3.5, demonstrating how BiAN achieves superior adaptability in counting tasks.

### 3.1 PRELIMINARY STUDY

In this section, we review the preliminary knowledge of cross-domain counting task. The objective of cross-domain counting is to train a network $\mathcal{N}$ that transfers the counting-relevant knowledge from source domain $D_s$ to $D_t$ with minimum joint decision error $\epsilon_U$. The network $\mathcal{N}$ process can be formulated as a Markov chain that $\mathcal{X} \xrightarrow{g} \mathcal{Z} \xrightarrow{f} \mathcal{Y}$. The error $\epsilon_U$ can be represented $\epsilon_U = \epsilon_{D_{s'}}(h) + \epsilon_{D_{t'}}(h)$, where $\epsilon_{D_{s'}}(h)$ and $\epsilon_{D_{t'}}(h)$ indicate the decision error on the transferred domains. The decision error $\epsilon$ can be represented as $\epsilon(h, f_i)$, where $h$ for hypothesis and $f_i^L$ for labeling function on the transferred domain (Zhao et al., 2019). The general DA interacts with domain-variant and domain-invariant features, which are $z_{var}$ and $z_{inv}$ respectively. The fundamental assumption is that $z_{var}$ does not influence the label $y$ (Kong et al., 2022). Specifically, the sketch of general DA can be represented as first identifying $z_{inv}$ and $z_{var}$, then processing $z_{inv}$ for recognition and migrating $z_{var}$ to the unified domain. Different from general DA approaches, the task of counting across domains introduces the concept of task-relevant factors $z_{task}$, which is domain-

variant but relevant to the results. Therefore, preserving $z_{task}$ is required for the stable counting adaptation process. In BiAN, we treat $z_{task}$ as contextual information between condition subsets and preserving it via conditional alignment and encourage network $\mathcal{N}_{BiAN}$ to maintain $z_{task}$. The definition of the elements can be represented as:

**Definition 1.** *Given domain-variant probability distributions $D_s$ and $D_t$ over an independent variable $\mathcal{X}$, which are $\mathcal{X}_s$ and $\mathcal{X}_t$ respectively. Let $g_s$ and $g_t$ be two reflections to project $\mathcal{X}_s$ and $\mathcal{X}_t$ to an overlapped feature domain $\mathcal{Z}_U$. The unified domain $\mathcal{Z}_U$ and label space $\mathcal{Y}$ can be represented as:*

$$\mathcal{Z}_s = g_s\left(\mathcal{X}_s\right),\ \mathcal{Z}_t = g_t\left(\mathcal{X}_t\right),$$
$$\mathcal{Z}_U = \mathcal{Z}_s \cup \mathcal{Z}_t, \mathcal{Z}_s \cap \mathcal{Z}_t \neq \varnothing.$$
$$\mathcal{Y}_s = f_s\left(\mathcal{Z}_s\right),\ \mathcal{Y}_t = f_t\left(\mathcal{Z}_t\right).$$

### 3.2 CONDITION ALIGNMENT

Within the framework of BiAN, we design conditional alignment with the following alignment strategy. It aims to independently align the conditional subsets $D_s^c = \{x_s^i, x_s^i \subseteq x^i \in D_s\}$ and $D_t^c = \{x_t^i, x_t^i \subseteq x^i \in D_t\}$ to maintain the distribution of contextual density information between conditions. It is straight to segment the entire feature into two condition subsets, which are objects of interest and background. In BiAN, there are two subsets to be aligned with minimal joint error.

In the following step, conditional alignment is adopted to operate the alignment depending on the segmentation results of images. The entire image $x$ is sent to recognize the relation between conditions. Then, the condition relation segments the entire image $x$ into object parts $x^f$ and background $x^b$. Lastly, these two subset features $z^f$ and $z^b$ can be obtained by feature extractor $g_{s/t}$ for conditional alignment. Specifically, the feature can be obtained by $z = g_{s/t}(x)$. Then, the object label prediction $\hat{y}$ can be obtained by $\hat{y} = f(z)$. If $(x, y) \in D_s$, we can further align the distribution convergence of $f(z)$ and $y$, which can be represented as:

$$g^* = \underset{g}{\arg\min}\, \mathcal{L}\left(f\left(g\left(\mathcal{X}_s\right)\right), \mathcal{Y}_s\right). \tag{1}$$

If $(x, y) \in D_t$, we still can obtain the pseudo $\hat{y}_t$ as the position-condition feature for the target domain $D_t$. After that point, $\hat{y}$ is applied as a mask indicator on $x$, then the image is divided into conditional subsets. Specifically, the mask can be generated from the predicted points of objects in $\hat{y}$ by extending range. The condition partitions $x^i$ can be represented as: $x = \bigcup_{i \in [f,b]} x^i\ \left(x^i \cap x^j = \varnothing, i \neq j\right)$. Then, the conditional partitions are sent to $g_i$ to obtain the conditional features $z_f$ and $z_b$. After that, we operate the alignment within the condition subset for all conditions included in the condition set $\mathcal{C}$. It means that every single alignment operation is only applied on $\bigcup_i z_i$. The operation can be represented as:

$$f^* = \underset{f}{\arg\min}\, d_{\mathcal{C}}\left(f\left(\mathcal{X}_s\right), f\left(\mathcal{X}_t\right)\right). \tag{2}$$

We suppose the combination of $f^*$ and $g^*$ are able to conditionally align the domain $D_s$ and $D_t$. According to Theorem 4, BiAN can achieve a lower joint decision error without being impacted by the conditional shift.

As for the specific model, we adopt SAU-Net (Guo et al., 2022) as the backbone of BiAN and modify it to make it capable of UDA counting tasks. Specifically, the components $g_{s/t}$ and $f$ source from the encoder and decoder in SAU-Net. To implement the aligning operation, the discriminator in DANN (Ganin & Lempitsky, 2015) is adopted as $f_d$ to fuse the domains by reversing the gradient during backpropagation.

### 3.3 CONDITION-CONSISTENT MECHANISM

In this section, we formally propose the Condition-consistent Mechanism (CM) to refine the pseudo labels in the target domain. Since the mask of the target domain is obtained via pseudo-labeling, it is essential to introduce CM to further enhance the self-supervised process. We suppose that partial distribution overlaps exist between domains. Thus, BiAN can learn to recognize part of target samples by leveraging knowledge from the source domain. After learning the distribution of objects, the network can directly segment the background and then learn the background feature. The obtained background feature distribution contrastively helps to learn about object features. Therefore, it is vital to maintain contextual information between condition subsets, which is our motivation for designing CM. In the conditional alignment process, the partitions sharing the same condition are sent to $g_{s/t}$ and $f$. Then, the results of $f(z^i)$ are expected to maintain as $y^i \in y$. Moreover, because $x^i \cap x^j = \varnothing$ when $i \neq j$, it is supposed that $y = concat\left(y^i\right)$, where $i \in f, b$. Specifically, we design a regressor $f_c$ which shares weights with $f$. For the evaluation of result consistency, we design a consistency loss, which can be represented as:

$$\hat{y}' = concat\left(f_c \circ g_t\left(x_t^f\right), f_c \circ g_t\left(x_t^b\right)\right), \tag{3}$$

$$\mathcal{L}_{CM} = \mathcal{L}\left(\hat{y}', f \circ g_t\left(x_t\right)\right), \tag{4}$$

where $f$ is the aforementioned regressor. We apply $MSE$ loss as $\mathcal{L}$. CM helps $f \circ g_t$ to transform different partial image information without annotation through minimizing $\mathcal{L}_{CM}$.

### 3.4 LOSS FUNCTIONS

In this section, we describe the loss function applied for training BiAN. The loss function can be divided into loss of source domain and loss of target domain. It can be represented as:

$$\mathcal{L} = \mathcal{L}_{source} + \mathcal{L}_{target} + \alpha\mathcal{L}_{CM}, \tag{5}$$

$$\mathcal{L}_{source} = \frac{\mathcal{L}_p\left(\hat{y}_s, y_s\right) + \mathcal{L}_p\left(\hat{y}_s^f, y_s\right) + \mathcal{L}_p\left(\hat{y}_s^b, \mathbf{0}\right)}{\mathcal{L}_d\left(\hat{c}_s^f, y_s^d\right) + \mathcal{L}_d\left(\hat{c}_s^b, y_s^d\right)}, \tag{6}$$

$$\mathcal{L}_{target} = \frac{\mathcal{L}_p\left(\hat{y}_t^b, \mathbf{0}\right)}{\mathcal{L}_d\left(\hat{c}_t^f, y_t^d\right) + \mathcal{L}_d\left(\hat{c}_t^b, y_t^d\right)}, \tag{7}$$

where $\hat{c}_{s/t}^* = \{0, 1\}$ is the output of $f_d(z_{s/t}^*)$, presenting the predication of which domain sample belonging to. And $y^d$ denotes the domain label of the sample. $\mathcal{L}_p$ is $MSE$ loss, $\mathcal{L}_d$ is applied reversed $NLL$ loss, maintaining $L_source$ positive. $\mathbf{0}$ in $\mathcal{L}_p$ presents background. The coefficient $\alpha$ presents the weight of $CM$ to balance the orders of magnitude with the rest of the loss elements. The employed $\mathcal{L}_d$ and $\mathcal{L}_p$ loss can be represented as:

$$\mathcal{L}_d\left(\hat{c}_i, y_i^d\right) = \frac{1}{N}\sum_{i=1}^{N} y_i^d \log(\hat{c}_i) + (1 - y_i^d)\log(1 - \hat{c}_i), \tag{8}$$

$$\mathcal{L}_p\left(\hat{y}_i, y_i\right) = \frac{1}{N}\sum_{i=1}^{N}(y_i - \hat{y}_i)^2, \tag{9}$$

where $y_i^d$ indicates the domain of the sample, $\hat{c}_i$ denotes the prediction of sample domain. $y_i$ is the ground-truth counting map of the sample, $\hat{y}_i$ is the prediction of counting map. $N$ indicates the amount of samples.

## 3.5 THEORETICAL ANALYSIS

In this section, we prove that the proposed BiAN can achieve a lower bound of joint decision error on both domains. First, the adaptation task can be represented as follows. For the source domain $D_s$ and the target domain $D_t$, our goal is searching the optimal decision hypothesis function $h^* = g \circ f$ to simultaneously reach the least joint decision loss $\lambda$ in all transferred domains. However, it has been proved that the unconditional alignment leads to the significant constraint of lowering the joint decision error, causing the burden of further increasing the adaptability of models (Zhao et al., 2019). Specifically, the goal of unconditional alignment can be represented as $\arg\min_h |d_{\mathcal{H}\Delta\mathcal{H}}(h(D), h(D'))|$. Zhao's paper (Zhao et al., 2019) has provided a comprehensive deduction that under the significantly large marginal difference between label space of domains, the joint decision error $|\epsilon_D(h^*, f) + \epsilon_{D'}(h^*, f')|$ has the lower bound as $|d_{JS}(\mathcal{Y}, \mathcal{Y}') - d_{JS}(D, D')|$. The constraint still holds while adopting the sophisticated unconditional transferring function. Therefore, we introduce a theorem of conditional adaptation and prove that it helps the adaptation model achieve lower joint decision error. We first introduce the definition of variables and symbols. Then, we describe our proposed theorem and provide the corresponding proof. Note that we provide the key definitions. The rest of symbols and variables in this paper follow the definitions in Ben-David's paper (Ben-David et al., 2009).

**Definition 2** (Divergence Measurement). *Given a hypothesis function $h$ and two domains $D$ and $D'$. Let $I$ be the identifying function. The divergence measurement between $D$ and $D'$ can be represented as:*

$$d_{JS}(D, D') = \frac{1}{2}d_{\mathcal{H}\Delta\mathcal{H}}\left(D, \frac{(D+D')}{2}\right) + \frac{1}{2}d_{\mathcal{H}\Delta\mathcal{H}}\left(D', \frac{(D+D')}{2}\right),$$

*where $d_{\mathcal{H}\Delta\mathcal{H}}(D, D') = 2sup|Pr_D[I(h)] - Pr_{D'}[I(h)]|$.*

**Definition 3** (Conditional Subset). *Given a domain probability distributions $D$ over $\mathcal{X}$. Let $\mathcal{C} = \{c_1, c_2, c_3, \ldots, c_k\}$ be a condition set of $D$. The conditional subset of $D$ is*

$$D = \bigcup_{i=1}^{k} D_i^c, i \neq j, D_i^c \cap D_j^c = \varnothing.$$

Specially, the condition set $\mathcal{C}$ denotes the attributes of partitions within samples (e.g. background and foreground in counting sample).

**Definition 4** (Conditional Divergence). *Given $D$ and $D'$ shared the same condition set $\mathcal{C}$, the conditional divergence can be defined as:*

$$d_{\mathcal{C}}(D, D') = \sum_{i \in [1,k]} d_{JS}(D_i^c, D_i'^c).$$

*If $d_{\mathcal{C}}(D, D') = 0$, $D$ and $D'$ are supposed as conditional aligned on condition $\mathcal{C}$.*

**Theorem 1** (Joint Error Lower Bound). *Based on the theorem proposed in (Zhao et al., 2019), combining the definition of the joint error $\epsilon_U = \epsilon_{\mathcal{Z}}(h) + \epsilon_{\mathcal{Z}'}(h)$ and the unified feature space $Z_U$, the corresponding lower bound can be rewritten as:*

$$\epsilon_U \geqslant \frac{1}{2}\left(d_{JS}(\mathcal{Y}, \mathcal{Y}') - d_{JS}(\mathcal{Z}, \mathcal{Z}')\right)^2.$$

**Lemma 2.** *Assume the label space $\mathcal{Y}$ of $D$ and $D'$ is discrete. If consider treat the label set as the condition set $\mathcal{C}$, the relation of $\mathcal{Y}$ and $\mathcal{Y}'$ can be presented as:*

$$d_{\mathcal{C}}(\mathcal{Y}, \mathcal{Y}') = 0.$$

According to the definition of $d_{JS}$ and $\mathcal{Y}, \mathcal{Y}'$, the labeling function is always consistent. So that $\mathcal{Y}, \mathcal{Y}'$ are always conditionally aligned when treating the label set as the condition set. Details of the proof can be found in Appendix A.3.

Table 1: Counting MAE and MSE on JHU-Crowd++ with labels "Stadium"(SD), "Street"(SR), "Snow"(SN) and "Fog/Haze"(FH). The best are highlighted in bold. DA: Domain Adaptation for short. DG: Domain Generalization for short.

| Method | DA | DG | SD→SR | | SR→SD | | SN→FH | | FH→SN | |
|---|---|---|---|---|---|---|---|---|---|---|
| | | | MAE ↓ | MSE ↓ | MAE ↓ | MSE ↓ | MAE ↓ | MSE ↓ | MAE ↓ | MSE ↓ |
| BL (Ma et al., 2019) | ✗ | ✗ | 42.1 | 79 | 262.7 | 1063.9 | 48.1 | 129.5 | 343.8 | 770.5 |
| MAN (Lin et al., 2022) | ✗ | ✗ | 45.1 | 79 | 246.1 | 950.8 | 38.1 | 68 | 445 | 979.3 |
| DAOT (Zhu et al., 2023) | ✔ | ✗ | 45.3 | 88 | 278.7 | 1624.3 | 42.3 | 73 | 151.6 | 273.9 |
| IBN (Pan et al., 2018) | ✗ | ✔ | 92.2 | 178 | 318.1 | 1420.4 | 109.7 | 267.7 | 491.8 | 1110.4 |
| SW (Pan et al., 2019) | ✗ | ✔ | 110.3 | 202.4 | 312.6 | 1072.4 | 131.5 | 306.6 | 381.3 | 825 |
| ISW (Choi et al., 2021) | ✗ | ✔ | 108.1 | 212.4 | 385.9 | 1464.8 | 151.6 | 365.7 | 276.6 | 439.8 |
| DCCUS (Du et al., 2023) | ✗ | ✔ | 90.4 | 194.1 | 258.1 | 1005.9 | 54.5 | 125.8 | 399.7 | 945 |
| MPCount (Peng & Chan, 2024) | ✗ | ✔ | 37.4 | 70.1 | 218.6 | 935.9 | 31.3 | **55** | 216.3 | 421.4 |
| BiAN (Ours) | ✔ | ✗ | **28.9** | **39.6** | **115.7** | **145.1** | **23.6** | 68.4 | **120.2** | **150.7** |

**Lemma 3.** *Given $D$ and $D'$ shared the discrete label space $\mathcal{Y}$ and set as the condition set $\mathcal{C}$, if $D_i^c \cap D' = D'^c$ the relation of the conditional subset of $D$ and the universal set of $D'$ can be presented as:*

$$d_{JS}\left(\mathcal{Y}_i^c, \mathcal{Y}_i'\right) = d_{JS}\left(D_i^c, D_i'^c\right).$$

According to the definition of $d_{JS}$ and the definition 4, the proof is obvious. The proof can be found in Appendix A.4.

**Theorem 4** (Conditional Alignment). *Given $D$ and $D'$ shared the discrete label space $\mathcal{Y}$ and set as the condition set $\mathcal{C}$, if $D$ and $D'$ is conditional aligned on label space $\mathcal{Y}$, then $d_{JS}\left(D, D'\right) = d_{JS}\left(\mathcal{Y}, \mathcal{Y}'\right)$.*

We present the proof in Appendix A.5. The deduction above shows that the joint error is bounded by the domain shift in both features and labels. Under reasonable assumptions, our proposed theorem presents a feasible approach to minimize the joint error by reducing the gap in both feature and label differences, rather than focusing solely on feature differences. The label difference is crucial in cross-domain counting scenarios, leading performance decay for significant label domain shift. We demonstrate that aligning feature partitions based on partition attributes preserves task-relevant information within the label distribution of the target domain, promoting the generalization of the model.

## 4 EXPERIMENTS AND RESULTS

### 4.1 EXPERIMENT SETTING

We conduct the experiments on eight domain combinations of different counting scenarios, including crowd counting and cell counting, to examine the adaptability of BiAN. For the crowd-counting task, the combination include "Stadium"(SD)"Street"(SR) and "Snow"(SN)-"Fog/Haze"(FH) within JHUCrowd++ (Sindagi et al., 2022),"Part A"(SHA)-"Part B"(SHB) within ShanghaiTech (Zhang et al., 2016), "Synthetic Fluorescence Microscopy"(VGG) (Xie et al., 2018b)-"Human subcutaneous adipose tissue"(ADI) dataset (Cohen et al., 2017), and "Dublin Cell Counting" (DCC) dataset (Marsden et al., 2018). The domain shift of crowd scenarios, including various weathers and densities, requires higher algorithm adaptability. For the cell counting task, the slight deviation of the cell amount of each image provides a comparative consistent density. However, various types of cells further challenge the performance of the model in the adaptability of scene presentation. The details of datasets and implementation are presented in Appendix A.6 and Appendix A.7. As to the evaluation metrics, we follow the previous works' setting. We employ only mean

Table 2: Counting MAE and MSE on crowd counting dataset ShanghaiTechA/B. The best are highlighted in bold. DA: Domain Adaptation for short.

| Methods | DA | SHB → SHA | | SHA → SHB | |
|---------|-----|-----------|-----------|-----------|-----------|
| | | MAE ↓ | MSE ↓ | MAE ↓ | MSE ↓ |
| CSRNet (Shi et al., 2019) | ✗ | 68.2 | 115.0 | 10.6 | 16.0 |
| KDM (Wan et al., 2022) | ✗ | 63.8 | 99.2 | 7.8 | 12.7 |
| UOT (Ma et al., 2021) | ✗ | 58.1 | 95.9 | 6.5 | 10.2 |
| STEERER (Han et al., 2023) | ✗ | 54.5 | 86.9 | 5.8 | 8.5 |
| CGNN (Zhang et al., 2024a) | ✗ | 61.1 | 97.8 | 7.7 | 13.0 |
| Cycle GAN (Zhu et al., 2017) | ✔ | 143.3 | 204.3 | 25.4 | 39.7 |
| SE CycleGAN (Wang et al., 2019b) | ✔ | 123.4 | 193.4 | 19.9 | 28.3 |
| BiTCC (Liu et al., 2020) | ✔ | 112.2 | 218.1 | 13.3 | 29.2 |
| LDG (Zhang et al., 2023) | ✔ | 118.5 | 190.1 | 14.2 | 25.2 |
| DGCC Du et al. (2023) | ✔ | 121.8 | 203.1 | 12.6 | 24.6 |
| SaKnD Xie et al. (2023a) | ✔ | 137.2 | 224.2 | 17.1 | 27.7 |
| CGNN-DA (Zhang et al., 2024a) | ✔ | 110.2 | 182.5 | 15.8 | 27.2 |
| BiAN (Ours) | ✔ | **42.3** | **53.0** | **5.7** | **7.2** |

absolute error (MAE) on cell counting as an evaluation metric, MAE and root mean squared error (MSE) on crowd-counting. Lower MAE and MSE indicate more precise counting results. This can be formulated as $MAE = \frac{1}{n} \sum_{i=1}^{n} |y_i - \hat{y}_i|$, $MSE = \sqrt{\frac{1}{n} \sum_{i=1}^{n} (y_i - \hat{y}_i)^2}$, which $y_i$ is the ground-truth count of the sample and $\hat{y}_i$ is the predication counts from model.

## 4.2 PERFORMANCE COMPARISON AND ANALYSIS

This section presents the results of our experiments on the baseline and the latest state-of-the-art models, categorized into two distinct scenarios: crowd counting and cell counting. The crowd-counting scenario presents a high-density variation situation. By applying established counting methodologies to datasets within both domains, we set the groundwork for assessing BiAN's advancements.

Table 3: Counting MAE on cell counting dataset combinations. The best are highlighted in bold. DA: Domain Adaptation for short.

| Methods | DA | VGG → ADI | VGG → DCC |
|---------|-----|-----------|-----------|
| | | MAE↓ | MAE↓ |
| Counting Focus (Shi et al., 2019) | ✗ | – | 3.2 |
| CCF (Jiang & Yu, 2020) | ✗ | 14.5 | – |
| AECC (Wang et al., 2021) | ✗ | 14.1 | 3.0 |
| SAU-Net (Guo et al., 2022) | ✗ | 14.2 | 3.0 |
| Two-Path Net (Jiang & Yu, 2021) | ✗ | 10.6 | – |
| MSCA-UNet (Qian et al., 2023) | ✗ | 9.8 | – |
| DTLCC (Wang, 2023) | ✔ | – | 3.0 |
| IDN (Liu et al., 2024) | ✔ | 11.1 | – |
| BiAN (Ours) | ✔ | **9.2** | **2.7** |

The experimental results, presented in Table 2 and Table 1, demonstrate that BiAN not only surpasses the latest state-of-the-art DA/DG methods but also achieves precision comparable to fully supervised methods in some combinations. These findings indicate that BiAN effectively adapts to cross-scene crowd counting scenarios. The results shows that BiAN

significantly improves counting precision over the latest state-of-the-art methods. These findings indicate that BiAN effectively adapts to cross-scene crowd counting scenarios.

Overall, the compared models cover supervised methods and adaptation methods. The counting approach of models includes density estimation, point-to-point prediction, and point-to-density prediction. In both cases, BiAN performs better than SOTA methods on counting tasks, demonstrating the effectiveness of our method. We present additional experiment analysis in Appendices, including the experiments on the setting of synthetic-real crowd counting (Appendix A.8), and qualitative investigation between condition feature consistency and counting results (Appendix A.8.2), and visualization results (Appendix A.10).

### 4.3 ABLATION STUDY

This section presents an ablation study to validate the effectiveness of our proposed method. We begin by removing all newly introduced mechanisms from the training process and implementing all variants across both counting tasks. The unconditional variant applies domain alignment to the entire condition partitions without aligning conditions independently, failing to retain the target task-relevant feature distribution It presents adaptation via style transfer. The variant *w/o* CM employs conditional alignment but excludes the CM module. The experimental results are shown in Table 4. It can be observed that the unconditional alignment domain adaptation only has limited adaptability. In adapting DCC, the unconditional-only variant performs worse than the existing adaptations due to the significant difference in the visual character of the cell between the two domains. It indicates that the marginal difference between the two label spaces might be significant. According to the findings in (Zhao et al., 2019), the model is hard to find the optimal combination of parameters to minimize joint errors.

In contrast, the samples in the crowd datasets share similar visual differences between partitions. Specifically, the scenes of the crowd are different. The difference between people and backgrounds is similar. In many instances, the background in crowd counting comprises other objects, leading to severe overlap situations compared to cell counting. In such cases, maintaining the margin distance between conditions is crucial. Incor-

Table 4: Ablation study evaluated by MAE.

| $MAE \downarrow$ | Unconditional | BiAN *w/o* CM | BiAN |
|---|---|---|---|
| VGG $\rightarrow$ ADI | 14.8 | 10.1 (-4.7) | 9.2 (-5.6) |
| VGG $\rightarrow$ DCC | 3.6 | 3.4 (-0.2) | 2.7 (-0.9) |
| GCC $\rightarrow$ UCF | 35.0 | 32.7 (-2.3) | 22.7 (-12.3) |
| SHB $\rightarrow$ SHA | 58.9 | 46.0 (-12.9) | 42.3 (-16.6) |

porating the CM module noticeably enhances the adaptability of BiAN, demonstrating its ability to maintain condition-independent partitions. The ablation experiments provide strong empirical evidence supporting the effectiveness of our proposed model's design, offering a persuasive explanation for its superior performance.

## 5 CONCLUSION

This paper proposes Binary Alignment Network (BiAN) to mitigate the domain shift with preserving task-relevant information in UDA counting tasks. The proposed conditional alignment enables the network to maintain the inter-object contextual distribution when aligning feature distribution across domains. Augmented by our Condition-consistent Mechanism (CM), the segment map can be further refined, enhancing the robustness of BiAN. We also provide a theoretical demonstration with reasonable assumptions of how conditional alignment beneficial to our task. We implement comprehensive experiments to demonstrate the effectiveness of BiAN.

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

# A  APPENDIX

## A.1  MORE RELATED WORK

In Unsupervised Domain Adaptation (UDA), there are common to adopt the component-wise alignment to align the feature distribution across domains. The most common method is to align the marginal distribution of the feature space (Jiang et al., 2022; Zhang et al., 2024b; Yao et al., 2023; Lopez-Rodriguez & Mikola-jczyk, 2022; Zhao & Wang, 2022; Zhou et al., 2023). MGA (Zhang et al., 2024b) designs category-level discriminator to align the distributions on the category-level. D-adapt (Jiang et al., 2022) deploys the bounding box alignment for mitigating the domain shift on object-level. Our method is actually different from these methods. Their goal is only to align the distribution of object-relevant features under the assumption that unconditional alignment can reduce the joint decision error all the time. It also assumes that the unconditional alignment will not violate the inter-object contextual distribution. Previous methods that align the entire image encourage the pseudo-label distributions of the source and target domains to converge, potentially overlooking significant gaps between the true label spaces due to inherent label shifts (Zhao et al., 2019). This issue is represented by the inequality:$d_{JS}(\mathcal{Y}, \mathcal{Y}') \geq d_{JS}(\mathcal{Y}, \hat{\mathcal{Y}}')$, where $d_{JS}$ denotes the Jensen-Shannon divergence between the label distributions $\mathcal{Y}$ (source domain), $\mathcal{Y}'$ (target domain), and $\hat{\mathcal{Y}}'$ (pseudo labels of the target domain). Although source domain labels are accessible, the inequality holds due to the unchangeable nature of the true label shift. According to Theorem 1, minimizing the estimated label difference encourages unsupervised domain adaptation methods to reduce $d_{JS}(\mathcal{D}, \mathcal{D}')$. However, combining this with the inequality leads to a higher lower bound of $\epsilon_U$, the target risk, due to the immutable label space difference. This theoretical insight explains why unconditional (global) alignment has limited adaptability in our context. And the consistency module enforces consistency between object and background features rather than maintaining features of ROI across domains, further enhancing the model's ability to capture essential density information. Technically, this module ensures that the model maintains consistency between object and background features, promoting accurate counting by preserving the relationships between objects and their surroundings. Theoretically, CM play a crucial role in maintaining the consistency of the feature space, implementing Definition 3. The proposed BiAN is designed to align the feature distribution across domains while preserving the inter-object information by maintaining the condition-independent partitions.

## A.2  TRAINING PROCESS

Here we provide the detailed training procedure of BiAN. The training process is shown in Algorithm A.2. The training process is similar to the standard training process of UDA. The difference is that we introduce the conditional alignment and the CM module to the training process. The images $(x_s, x_t)$ from the source and target domain are fed into the model $h = g_{s/t} \circ f$. The model firstly predicts the counting results $\hat{y}_s$ and $\hat{y}_t$. Then model segments the images into foreground and background using the predicted results, obtaining $(x_s^f, x_s^b)$ and $(x_t^f, x_t^b)$. The model then predicts the conditional results $(\hat{y}_s^f, \hat{y}_s^b)$ and $(\hat{y}_t^f, \hat{y}_t^b)$ for the foreground and background. The conditional domain loss $\mathcal{L}_d$ is calculated between the conditional results. The pixel loss $\mathcal{L}_p$ is calculated between the predicted results and the ground truth. The source loss $\mathcal{L}_{\text{source}}$ and the target loss $\mathcal{L}_{\text{target}}$ are calculated. The CM loss $\mathcal{L}_{\text{CM}}$ is calculated between the conditional results and the predicted results. The sum up loss $\mathcal{L}$ is calculated. The gradient of the loss is calculated and the model is updated.

**Algorithm 1** Training Procedure of BiAN

**Require:** Source dataset $\mathcal{D}_s$, Target dataset $\mathcal{D}_t$, Model parameters $\theta$, Learning rate $\eta$, Epochs $E$
**Ensure:** Trained model parameters $\theta$
1: **for** epoch = 1 **to** $E$ **do**
2:  Shuffle dataset $\mathcal{D}_s$, $\mathcal{D}_t$
3:  **for** each batch $B = (x_s, x_t)$ **do**
4:   Compute predictions $\hat{y}_s$, $\hat{y}_t$ using model with parameters $\theta$
5:   Get $(x_s^f, x_s^b)$, $(x_t^f, x_t^b)$ segmenting $x_s$, $x_t$ with $\hat{y}_s$, $\hat{y}_t$
6:   Compute conditional predictions $(\hat{y}_s^f, \hat{y}_s^b)$, $(\hat{y}_t^f, \hat{y}_t^b)$ using model with parameters $\theta$
7:   Compute conditional domain loss $\mathcal{L}_d$ between $(\hat{c}_s^f, \hat{c}_t^f)$ and $(\hat{c}_s^b, \hat{c}_t^b)$
8:   Calculate pixel loss $\mathcal{L}_p$ between $\hat{y}_s$ and ground truth $y_s$
9:   Calculate $\mathcal{L}_{source}$ and $\mathcal{L}_{target}$
10:   Calculate CM loss $\mathcal{L}_{CM}$ between $(\hat{y}_t^f, \hat{y}_t^b)$ and $\hat{y}_t$
11:   Calculate sum up loss $\mathcal{L} = \mathcal{L}_{source} + \mathcal{L}_{target} + \lambda\mathcal{L}_{CM}$
12:   Reverse the gradient of $\mathcal{L}_d$ then compute gradient $\nabla_\theta\mathcal{L}$
13:   Update parameters: $\theta \leftarrow \theta - \eta \cdot \nabla_\theta\mathcal{L}$
14:  **end for**
15:  **if** early stopping condition is met **then**
16:   **break**
17:  **end if**
18: **end for**
19: **return** $\theta$

## A.3 PROOF OF LEMMA 2

*Proof.*
$$d_{\mathcal{C}}(\mathcal{Y}, \mathcal{Y}') = \sum_{i \in [1,k]} d_{JS}(\mathcal{Y}_i, \mathcal{Y}_i'),$$
According to the definition, the samples within the condition subsets share the same label. So that, according to the previous definition of $d_{JS}(D, D')$, we have for every $i \in [1, k]$:
$$d_{\mathcal{C}}(\mathcal{Y}, \mathcal{Y}') = 0,$$
□

## A.4 PROOF OF LEMMA 3

*Proof.* According to Definition 3, we have
$$D_i^c \cap D_j'^c = \varnothing, i \neq j.$$

This implies that the subsets $D_i^c$ and $D_j'^c$ are disjoint whenever $i \neq j$. Since $D'$ can be expressed as the union of all such $D_j'^c$, for any $x \in D'$, it specifically lies in one of these subsets $D_i'^c$ if $x$ also belongs to $D_i^c$. Therefore, we have:
$$x \in D_i^c \cap D' \implies x \in D_i^c \cap D_i'^c.$$
Since $D_i^c \cap D' = D_i^c \cap D_i'^c$, the Jensen-Shannon divergence calculation between $D_i^c$ and $D'$ simplifies to:
$$d_{JS}(D_i^c, D') = d_{JS}(D_i^c, D_i'^c).$$
This holds because the overlap between $D_i^c$ and $D'$ is exactly $D_i^c$ and $D_i'^c$, thus limiting the scope of the divergence calculation to these intersecting subsets. □

### A.5 PROOF OF CONDITIONAL ALIGNMENT

*Proof.* The situation of the large marginal difference on label space can be represented as follows. Given condition set $\mathcal{Y} = \{c_1, c_2, c_3, \ldots, c_k\}$, for any $i \in [1, k]$, we have:

$$Y_i^c \cap Y' = Y_j'^c, i \neq j.$$

Without loss of generality, we suppose $j = i + 1$, so that we have:

$$d_{JS}(Y, Y') = \sum_{i=1}^k d_{JS}(Y_i^c, Y_j'^c).$$

Specially, we set $Y_{k+1}'^c = Y_1'^c$.

We have conditional aligned domains $D$ and $D'$, which can be represented as:

$$d_{\mathcal{C}}(D, D') = 0.$$

Therefore, for any $i \in [1, k]$:

$$d_{JS}(D_i^c, D_i'^c) = 0.$$

We have conditional aligned $\mathcal{Y}$ and $\mathcal{Y}'$, so it can instantly have:

$$d_{JS}(D_i'^c, Y_i'^c) = 0.$$

Combining the equations above, we have:

$$d_{JS}(D_i^c, Y_i'^c) = 0.$$

According to Lemma 2, we have:

$$d_{JS}(Y_i^c, Y_j'^c) = d_{JS}(D_i^c, Y_j^c) = d_{JS}(D_i^c, D_j^c) = d_{JS}(D_i^c, D_j'^c).$$

It is possible to find an order of sorting the $D_i^c$ and $D_i'^c$, so that the JS-convergence between $D$ and $D'$ can be:

$$d_{JS}(D, D') = \sum_{i=1}^k d_{JS}(D_i^c, D_j'^c).$$

Specifically, we set $D_{k+1}'^c = D_1'^c$. To this end, combining the equations above, we have:

$$d_{JS}(D, D') = d_{JS}(Y, Y').$$

$\square$

### A.6 DATASET DETAILS

In this section, we will provide details about the dataset we implemented in our experiments, including cell counting datasets and crowd counting datasets. Example visualization is shown as Figure 3.

For the crowd-counting task, the datasets include GTA5 Crowd Counting (GCC) (Wang et al., 2019b), UCF-QNRF (UCF) (Idrees et al., 2018), ShanghaiTech (SHA & SHB) (Zhang et al., 2016), and JHU-Crowd++Sindagi et al. (2022). The details of the crowd dataset are shown as follows:

- GCC (Wang et al., 2019b) is generated from multiple crowd scenes in Grand Theft Auto V, a video game, with 15,210 samples. The image size is 1920×1080 pixels. The synthetic environment contains multiple times of the day, seven types of weather, and diverse scenes (e.g. beach, street, and other common public scenes.). It provides various simulations of real-world scenes. The average of crowded count for each image is 500, with the highest count of 4000 and lowest count of zero.

- UCF (Idrees et al., 2018) is a large-scale dataset that contains 1535 high solution images with considerable crowd variation. The images are obtained from the Web by multiple platforms. So, the resolutions are highly dynamic. The average density of images is 1000 counts but with a standard deviation 7605.14.

- The ShanghaiTech (Zhang et al., 2016) dataset consists of parts A and B, containing 482 and 716 samples, respectively. Part A (SHA) is obtained from the Web with dynamic resolutions. The mean of counts per image is 541, with a standard deviation of 504. Part B (SHB) is retrieved from the security monitoring cameras on busy streets with fixed resolutions. The mean of counts per image is 122, with a standard deviation 93.

- The JHUCrowd++ (Sindagi et al., 2022) dataset consists of 4,372 images with detailed annotations, totaling approximately 1.51 million instances. The images are collected from diverse sources, including the web and surveillance cameras, featuring varying resolutions and perspectives. The dataset captures a wide range of crowd densities, from sparse to extremely dense scenes. The mean count per image is approximately 346, with a standard deviation of 1,094, indicating significant variability in crowd counts across the dataset.

The environments of the crowd datasets, including various weathers and scenes, are among the most challenging issues to handle in crowd counting. It requires algorithms with higher adaptability to handle it. Overall, the selection of datasets covers a sufficient variety of environments and scenes. In the following experiments, we examine the transferability of the BiAN by evaluating its performance in transferring features between the domains from the datasets shown above.

For the cell counting task, the datasets include three public benchmarks: synthetic fluorescence microscopy (VGG) dataset (Xie et al., 2018b), human subcutaneous adipose tissue (ADI) dataset (Cohen et al., 2017), and Dublin Cell Counting (DCC) dataset. The details of the cell dataset are shown as follows:

- VGG (Xie et al., 2018b) is a synthetic microscopy cell image dataset with 200 samples. It simulates bacterial cells from fluorescence-light microscopy at various focal distances. The size of microscopy images is maintained as $256 \times 256$ pixels. The cell amount of VGG for each image is $174 \pm 64$.

- DCC (Marsden et al., 2018) dataset is built with 177 samples from various categories of cells from real cases, including embryonic mice stem cells, human lung adenocarcinoma, and monocytes. The image size ranges from $306 \times 322$ pixels to $798 \times 788$ pixels, due to obtained via dynamic zoom scope. Moreover, the cell amount for each image is $34 \pm 21$, intended to increase the variation of the dataset.

- ADI (Cohen et al., 2017) is constructed from Genotype Tissue Expression Consortium (Lonsdale et al., 2013) with densely packed adipocyte cells from real cases. The dataset is built from 200 images. The image size is $150 \times 150$ pixels. The cell amount for each image is $165 \pm 44$.

The slight deviation of the cell amount of each image provides a relative consistency in cell density. Various types of cells further challenge the performance of the model in the adaptability of scene presentation.

### A.7   EXPERIMENT IMPLEMENTATION DETAILS

We choose the Adam optimizer with decoupled weight decay. The learning rate for the optimizer is set to 1e-6, and the weight decay rate is 1e-4. For the learning rate, we use a step learning rate scheduler with a 10-epoch step to lower the learning rate by 0.1 for every step. To handle the limitation of GPU memory, we resize cell images to $128 \times 128$ pixels and crowd images to $96 \times 96$ pixels. Notably adopting the annotated counts before resizing the images to maintain the ground truth unaffected by squeezing. The coefficient $\alpha$ of CM loss is set to 100. Moreover, we apply the training scalar on the annotations to enhance the numeric difference. The scalar for VGG and ADI is 100. For DCC and all applied crowd datasets, it is set as 500, respectively. BiAN is fully implemented in PyTorch, running on a single NVIDIA RTX 3090 with a single Intel® Core™ i7-10700 CPU @ 2.90GHz.

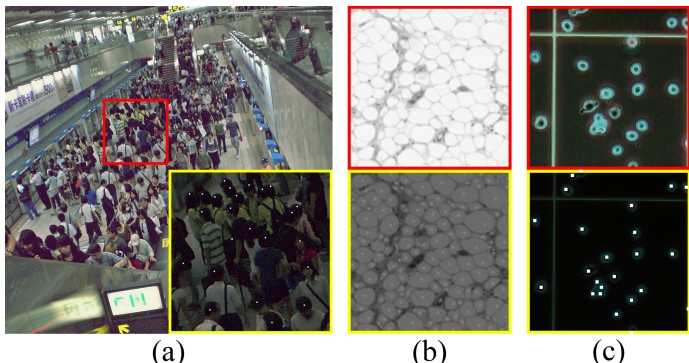

$$(a) \qquad\qquad (b) \qquad\qquad (c)$$

Figure 3: Object counting scenarios: (a) public security monitoring; (b) medical pathological analysis; (c) biological experiment.

## A.8 ADDITIONAL EXPERIMENT ANALYSIS

### A.8.1 SOURCE ON SYNTHETIC CROWD DATASET

Migrating from the source synthetic dataset to a real-world dataset is a practical approach to handling insufficient data annotation issues. To validate BiAN performance on such condition, we conduct the experiments with the setting of GCC (*source*) and UCF (*target*). Specifically, we have to resize the input to a smaller size (128×128 px) due to memory limitation. We have taken reasonable measures to preserve the information. The results still show that BiAn outperforms SOTA methods. But due to no guarantee on lost information, the experiment results only can be quantitatively referred.

Table 5: Counting MAE and MSE on crowd counting task from synthetic source. The best are highlighted in bold. DA: Domain Adaptation for short.

| Methods | DA | GCC → UCF | |
| --- | --- | --- | --- |
| | | MAE ↓ | MSE ↓ |
| KDMG (Wan et al., 2022) | ✗ | 99.5 | 173.0 |
| UOT (Ma et al., 2021) | ✗ | 83.3 | 142.3 |
| STEERER (Han et al., 2023) | ✗ | 74.3 | 128.3 |
| Cycle GAN (Zhu et al., 2017) | ✔ | 257.3 | 400.6 |
| SE CycleGAN (Wang et al., 2019b) | ✔ | 230.4 | 384.5 |
| BiAN (Ours) | ✔ | **22.7** | 28.4 |

### A.8.2 RELEVANCE ANALYSIS BETWEEN CONSISTENCY AND COUNTING RESULTS

In this section, we further demonstrate the proposed Condition-Consistency Mechanism (CM), which benefits from reliable counting when there is a lack of precise annotation during the adaptation process. We plot the curves presenting the tendency of MAE on the validation set and uncertainty during the training period. Specifically, the uncertainty index is calculated by the normalized CM loss $NORM(\mathcal{L}_{CM})$, indicating how inconsistent the features of assembling conditions and entire ones are. It can be observed that the counting performance, which is inversely proportional to the MAE value, is promoted when the uncertainty index

decays. Combined with the results in experiment results in Section 4.3, it can validate that the assumption on disjoint condition subsets is necessary in BiAN and conditional alignment framework.

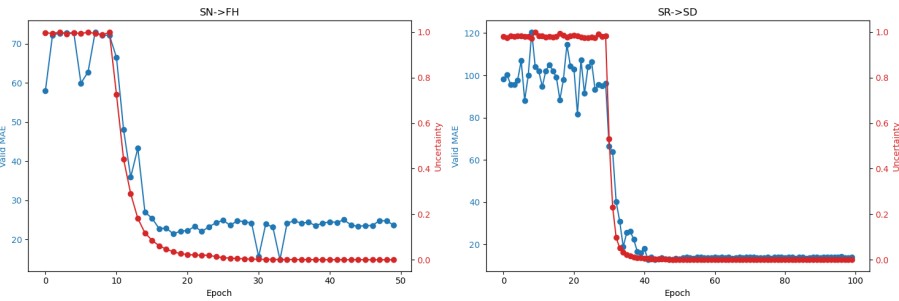

Figure 4: The tendency of validation counting MAE and the consistency on two domain combinations.

## A.9 MODEL ARCHITECTURE OF BiAN

In this section, we present the architectural details of our proposed BiAN. As shown in Figure 2, the architecture can be divided into domain-specific feature extractors ($g_{S/T}$), density map regression layers ($f$), domain discriminator ($f_d$), condition-consistent layers ($f_c$). The total parameters amount of BiAN is 70,755,271 with an estimated model size of 2783.41 MB.

Table 6: Architecture of feature extractor in BiAN.

| Layer (Type: Depth-Idx) | Output Shape | Parameters (#) |
|---|---|---|
| Conv2d: 3-1 | [32, 256, 256] | 384 |
| Conv2d: 3-2 | [32, 256, 256] | 9,312 |
| MaxPool2d: 3-3 | [32, 128, 128] | – |
| Conv2d: 3-4 | [64, 128, 128] | 18,624 |
| Conv2d: 3-5 | [64, 128, 128] | 37,056 |
| MaxPool2d: 3-6 | [64, 64, 64] | – |
| Conv2d: 3-7 | [128, 64, 64] | 74,112 |
| Conv2d: 3-8 | [128, 64, 64] | 147,840 |
| MaxPool2d: 3-9 | [128, 32, 32] | – |
| Conv2d: 3-10 | [256, 32, 32] | 295,680 |
| Conv2d: 3-11 | [256, 32, 32] | 590,592 |
| SelfAttention: 3-12 | [256, 32, 32] | 263,424 |

The domain-specific feature extractors ($g_{s/t}$), detailed in Appendix A.9, are responsible for capturing relevant features from the input data in both source and target domains. Specifically, the input is resized as 256×256 px. And the network arguments are independent among $g_s$ and $g_t$, but same architecture for similar feature retrieval.

The density map regression layers ($f$), detailed in Table 7, are designed to predict density maps from the extracted features. Specifically, it expands the channels of features by deconv operations and estimates the density. The output size is input-alike but only one channel, which is shown as resized 256×256 px. And the $f_c$ shares weight and architecture with $f$ for preparing partial results map for CM validation.

The domain discriminator ($f_d$), as described in Table 8, aims to align the domain distribution shift of features by broadcasting inverse gradient. The output is the binary label.

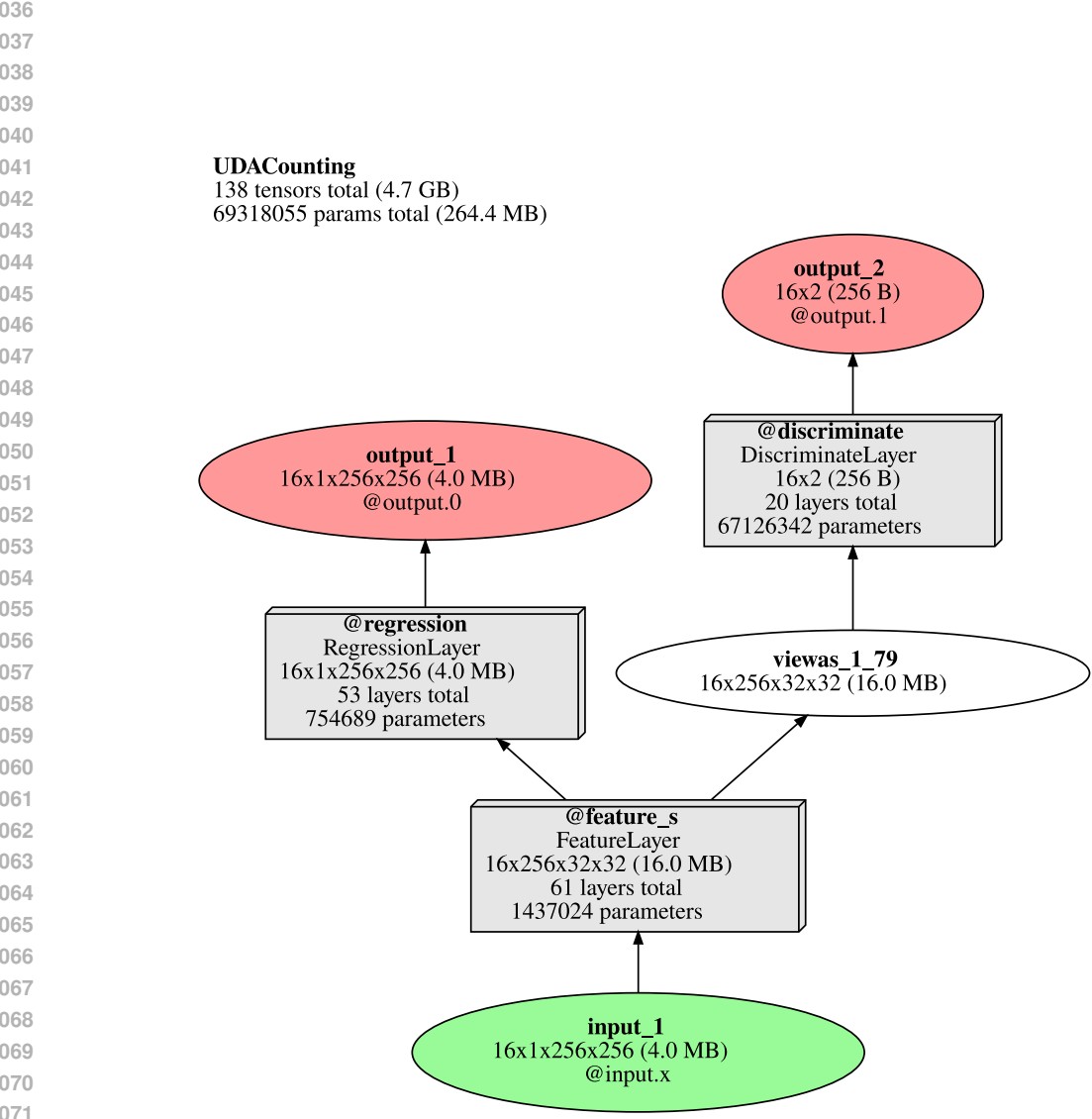

Figure 5: The network architecture visualization of BiAN. Note: The sample input is $16\times1\times256\times256$ px. Then estimated allocated memory is 4.7 GiB.

Table 7: Architecture of regression layers in BiAN.

| Layer (Type: Depth-Idx) | Output Shape | Parameters (#) |
|---|---|---|
| Deconv2d: 3-13 | [128, 64, 64] | 131,456 |
| Conv2d: 3-14 | [128, 64, 64] | 295,296 |
| Conv2d: 3-15 | [128, 64, 64] | 147,840 |
| Deconv2d: 3-16 | [64, 128, 128] | 32,960 |
| Conv2d: 3-17 | [64, 128, 128] | 73,920 |
| Conv2d: 3-18 | [64, 128, 128] | 37,056 |
| Deconv2d: 3-19 | [32, 256, 256] | 8,288 |
| Conv2d: 3-20 | [32, 256, 256] | 18,528 |
| Conv2d: 3-21 | [32, 256, 256] | 9,312 |
| Conv2d: 3-22 | [1, 256, 256] | 33 |

Table 8: Architecture of domain discriminate layers in BiAN.

| Layer (Type: Depth-Idx) | Output Shape | Parameters (#) |
|---|---|---|
| Linear2d: 3-23 | [256] | 67,109,632 |
| Linear2d: 3-24 | [64] | 16,576 |
| Dropout1d: 3-25 | [64] | – |
| Linear2d: 3-26 | [2] | 134 |
| Softmax: 3-27 | [2] | – |

## A.10 VISUALIZATION

In this section, we present the visual results of BiAN in the counting task experiments. As shown in Figure 7, we randomly select two samples from every cross-domain adaptation. In the visualization figure, we mark the inaccurate counts in the samples. The low-density samples can be counted in precise amounts, and the localization is also accurate. However, in microscopy cell images, cells of an overlapped or abnormal size are not fully recognized. The cell-alike objects (e.g. bubbles) easily distract the model recognition, especially in the DCC cell images. The conditional alignment mechanism enables BiAN to recognize distinguishing features of cells. As for the crowd counting task, human main characters are important cues to lead the model to marks. In contrast, the characters of hidden persons are easily missed targets. The results show that BiAN is able to retrieve the partial features of humans. It results in significant performance improvements. Overall, the visualization demonstrates the proposed model's recognition ability and learning of the visual representation of counting targets.

## A.11 LIMITATIONS AND FUTURE WORK

This paper has several limitations that can be further investigated and improved. First, the lower bound of the aforementioned joint error is not the tightest (Zhao et al., 2019). It means the tightest lower bound of joint error might be lower than the loss bound mentioned above. However, this does not explicitly result in a performance drop. Second, the conditions are the label categories in BiAN, however Theorem 4 can fit more conditions. Regarding the designed model BiAN, we observed that there exists limited precision in recognizing and localizing, and counting minor objects. This aspect can be further improved in addition to the proposed CM. Nevertheless, our work emphasizes the importance of matching the conditions during the adaptation process and provides a promising direction for future research.

Figure 6: Density map visualization. Randomly selected two high-density samples from JHUCrowd++. The left ones are predictions, the right ones are labeled density maps.

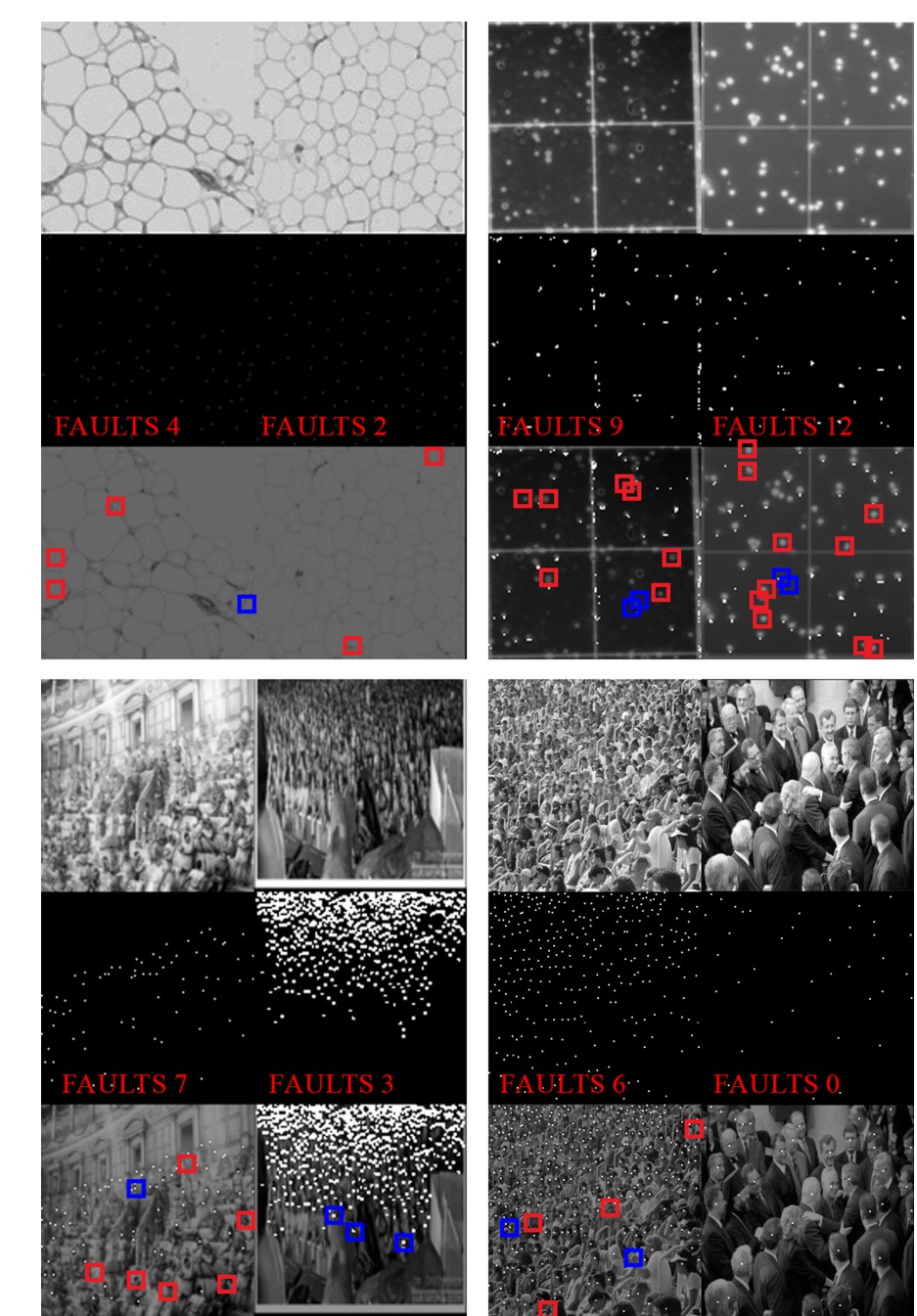

Figure 7: Dot map visualization. Randomly selected eight low-density samples from two adaptation tasks. From left to right, the samples are from ADI, DCC, UCF, SHB. The red mark indicates the miss count. The blue mark indicates the duplicated count.

