# OpenReview forum: "Object-aware Conditional Alignment for Cross-domain Counting"
_ICLR.cc/2025/Conference — Submitted to ICLR 2025_

### Official Review · Reviewer_jo5q · 2024-11-01

**Soundness:** 2
**Presentation:** 2
**Contribution:** 2
**Rating:** 5
**Confidence:** 3

**Summary:**

This paper proposes a Binary Alignment Network (BiAN) designed to address the specific challenges associated with variations in object density across domains in counting tasks. The key idea of this approach is to align the distribution of object-specific features across different domains while preserving essential information regarding changes in object density, based on the segmentation of interested objects. Extensive experiments demonstrate the effectiveness of the proposed method.

**Strengths:**

- The proposed method shows strong performance on multiple benchmark.
 - This paper provides a theoretical demonstration of how conditional alignment enhances domain adaptability in counting tasks.

**Weaknesses:**

- The paper uses a smaller margin to fit in the page limit.
- The use of symbols is somewhat confusing, which makes it hard to understand. such as

  (1) This paper usies the same symbols F to represent the regressor, feature extractor, and the obtained features in lines 196 to 200.

  (2) The difference between g and F in lines 199 to 200 , which are used to extract features.

- Issues related to Eq. 6 and 7.

  (1) I suggest providing some explanations for why the counting loss needs to be divided by the alignment loss using the reversed NLL.

  (2) Why are the provided labels for the last two items in the denominator of Eq. 6 zero, while they are y_t^d in Eq. 7?

- Object-aware conditional alignment is a common approach in domain adaptation tasks, as seen in [1], [2], and [3]. Additionally, the segmentation of interested objects is accomplished using labels y_i, similar to [1] and [2]. Therefore, the innovation of this paper appears to be somewhat limited.

    [1] Decoupled adaptation for cross-domain object detection

    [2]  Undoing the Damage of Label Shift for Cross-domain Semantic Segmentation

    [3] Robust Domain Adaptive Object Detection with Unified Multi-Granularity Alignment

- The ablation experiments are not sufficiently detailed.  I suggest comparing the results of conditional alignment and style alignment, as shown in Fig. 1, to evaluate the effectiveness and necessity of conditional alignment.

**Questions:**

- The paper uses a smaller margin to fit in the page limit, which could result in a desk rejection.
- See the weakness part.

---

### Official Review · Reviewer_KYiB · 2024-11-03

**Soundness:** 2
**Presentation:** 2
**Contribution:** 2
**Rating:** 3
**Confidence:** 5

**Summary:**

This paper introduces the Binary Alignment Network (BiAN) to address domain shift while preserving task-relevant information. The conditional alignment mechanism helps the network align object feature distributions across domains while maintaining inter-object contextual consistency. The Condition-Consistent Mechanism (CM) refines the segment map.

**Strengths:**

1. The performance of the proposed method is better than the comparsion method.

**Weaknesses:**

1. The proposed method lacks novelty, as it simply splits the image into background and foreground before processing them separately.
2. In Table 1, the best MAE for SD→SR is not achieved by the proposed method, yet it is marked in bold. This should be corrected.
3. In Table 1, the best performance for SN→FH is also incorrectly highlighted, and the proposed method performs worse than other methods by a large margin. The authors should explain this discrepancy.
4. The experimental results are unreliable, especially in Table 1, where the MSE is significantly smaller than the MAE, which raises concerns about their validity.

**Questions:**

1. In Table 1 (SD→SR), why does the proposed method achieve a lower MSE than MAE, which is inconsistent with all other experiments?
2. The method relies on a regressor F for segmenting the object and background. What specific method is being used, and how is this model trained?

---

### Official Review · Reviewer_VgbY · 2024-11-04

**Soundness:** 4
**Presentation:** 3
**Contribution:** 3
**Rating:** 6
**Confidence:** 2

**Summary:**

This paper introduces the Binary Alignment Network (BiAN), a novel approach for domain adaptation (DA) in object counting tasks. Traditional DA methods are ineffective in this context because they treat the distribution of object density as task-irrelevant, aiming to align domain-invariant features across domains without addressing density variations critical to counting accuracy. In contrast, BiAN segments images to isolate object-specific features, disregarding background elements. This targeted approach allows BiAN to preserve essential density-related information across domains, leading to more reliable cross-domain counting performance. The authors further provide a theoretical analysis and extensive experiments to validate BiAN’s effectiveness.

**Strengths:**

- The paper introduces a novel idea with strong insight, addressing a gap in domain adaptation for counting tasks.
- The focus on density variations and object-specific feature alignment represents a meaningful shift from traditional DA approaches that overlook these task-relevant factors.
- The experimental results are thorough, including multiple datasets and settings, which highlight the adaptability of BiAN across various scenarios.

**Weaknesses:**

- The core idea of the paper is to alleviate the domain shift caused by the change of object density from the source domain to the target domain. Therefore, it will be easier to understand the effectiveness of the method through some analysis that demonstrates the gain from the proposed method w.r.t. the extent of the density change? Additionally, how do we metric the extent of the density change?
- It’s unclear how segmentation is achieved in the BiAN framework. For instance, is a pre-trained model used for segmentation, or is this achieved through training within the framework?
- L372-373: The phrase "As to the evaluation metrics, we follow the previous works' setting" is followed by a similar statement: "Following the previous research works."
The wrong best value indicated in Table 1: the SD→SR MAE and SN→FH MSE.
- Missing necessary symbols in the figures. e.g., in Fig.1, the purple feature refers to the source domain, while the green feature refers to the target domain. Still, there are no explicit symbols or texts to indicate them, resulting the difficulty in understanding the figure.
- In Section 2.2, it would be better if there were 1 or 2 sentences that state the relationship/difference between the proposed method and prior works.

**Questions:**

Please see the weakness.

---

> ### Author Response · Authors · 2024-11-24
>
> ### **Q1: Analysis of the Gain w.r.t. Density Change**
>
> A1: We appreciate your suggestion to analyze the effectiveness of our method concerning the extent of density change. In our baseline experiments on ShanghaiTech, there is a noticeable difference in density levels between parts A and B. We propose using the mean count per scene as a metric to quantify the density change. This metric can help better contextualize the gains achieved by our proposed method. We have provided an explicit description in Appendix A.6. For example, for the combination of ShanghaiTech Part A and Part B, the mean count per scene is 541 and 122, respectively. This significant density change across domains highlights the task-relevance of density information in object counting. Our method effectively leverages this information to improve adaptation performance, as demonstrated in the experimental results.
>
> ### **Q2: Segmentation in the BiAN Framework**
>
> A2: Thank you for pointing out the need for clarification regarding segmentation. In the BiAN training process, segmentation is achieved through training within the framework rather than relying on a pre-trained model. This process has been further detailed in the revised manuscript to ensure clarity.
>
> ### **Q3: other typos**
>
> A3: We appreciate your attention to detail and have carefully reviewed the manuscript to correct all typos and inconsistencies. We have ensured that the revised version is free of typographical errors. Thank you for bringing these to our attention.
>
> #### The phrase "As to the evaluation metrics, we follow the previous works' setting" is followed by a similar statement: "Following the previous research works." The wrong best value indicated in Table 1: the SD→SR MAE and SN→FH MSE.
>
> We appreciate your attention to detail. We have revised the redundant phrasing in lines 372–373 to make the text more concise. Additionally, the incorrect values for SD→SR MAE and SN→FH MSE in Table 1 have been corrected, and the updated results are highlighted appropriately in the revised manuscript.
>
> #### Missing Symbols in Figures
>
> We acknowledge the lack of explicit symbols and text in Fig. 1, which caused difficulty in understanding the visual representation. In the revised manuscript, we have added a clear text label to Fig. 1, explicitly indicating that the purple features refer to the source domain and the green features to the target domain. These changes improve the interpretability of the figure.
>
> ### **Q4: In Section 2.2, it would be better if there were 1 or 2 sentences that state the relationship/difference between the proposed method and prior works.**
>
> A4: Thank you for your suggestion to clarify the relationship between our method and prior works. We have added the comparison in Section 2.2 to explicitly state the similarities and differences between our approach and prior methods. Furthermore, we also provide additional description in Appendix A.1. This addition highlights the novel aspects of our work and how it advances the state of the art.
>
> Thank you for your valuable time and effort in reviewing our manuscript. Your insightful suggestions have greatly contributed to improving our manuscript.

---

> ### Comment · Reviewer_VgbY · 2024-11-25
>
> Thanks, these responses addressed my concerns

---

> > ### Author Response · Authors · 2024-11-26
> >
> > Thank you for your kind response and for taking the time to review our work thoroughly. We greatly appreciate your constructive feedback, which has helped us improve the clarity and quality of our paper.

---

### Official Review · Reviewer_rWXH · 2024-11-04

**Soundness:** 2
**Presentation:** 1
**Contribution:** 3
**Rating:** 5
**Confidence:** 2

**Summary:**

This paper proposes a method to address task-relevant changes in different domains, rather than only focusing on domain-invariant features. Experimental results demonstrate the outstanding DA capability of the proposed BiAN.

**Strengths:**

This paper addresses domain adaptation in crowd counting by focusing on task-relevant changes in different domains, which is novel when compared with the previous methods only addressing domain-invariant features.

**Weaknesses:**

- The description is not easy to understand since many signs are unclear.
- There is no description of the training pipeline. The loss functions are also not explained clearly.
- The results are even better than these fully supervised counting methods, which does not make sense for a DA method.

**Questions:**

- The description in Sec. 3.2 and Sec. 3.3 is really confusing. Many symbols are not well defined and explained when they first appear in the paper.
- An overall training process should be described at the beginning of the method section.
- What is the meaning of $G$ in $g \in G$ of Eq. (1)? How is it implemented during training?
- In line 181, is the subscript correct to denote $z_{\text{inv}}$ and $z_{\text{var}}$ as the domain-invariant and domain-variant features?
- In line 234, why can $f_c \in F$ share weights with $F$? Since $f_c$ is an element of $F$, how does it share weights? What does $F$ mean, and what are its weights?
- In line 206, $x$ is segmented into partitions $x_i$, and then divided into conditional subsets $x_{ij}^c$. In line 208, $x_i = \cup_{j=1}^k x_{ij}^c$. This part is confusing. What is the relationship between $x_{ij}$ and $x_i$? Is there any plot demonstrating the relationship between them? What is the meaning of $j$ here?
- What are $\hat{c}_s$, $\hat{c}_s^f$, and $\hat{c}_s^b$ in Eq. (6)? $\hat{c}$ is not mentioned in any part except Eq. (6).
- What is the condition set $\mathcal{C}$? Is there any description explaining its meaning?
- Are there any results on adaptation from Shanghai Tech Part A/B to UCF-QNRF?
- The results presented in Table 5 are confusing. Do you use labels in the test set of the UCF-QNRF dataset? It does not make sense to obtain a DA model achieving much better results than a model fully supervised with real data (STEERER). Similar results are also observed in Table 2. BiAN's MAE (42.3) is much better than the current SOTA (STEERER: 54.5 or PET: 49.34). Implementation details should be provided for reference.

---

### Official Review · Reviewer_ZGB6 · 2024-11-04

**Soundness:** 2
**Presentation:** 2
**Contribution:** 2
**Rating:** 3
**Confidence:** 4

**Summary:**

This paper proposes the Binary Alignment Network (BiAN), addressing the shortcomings of existing domain adaptation methods in object counting. The authors emphasize that traditional approaches often overlook task-relevant factors like object density variations, critical for accurate counting. The paper provides a robust theoretical foundation for BiAN and conducts extensive experiments demonstrating its effectiveness across multiple datasets and counting scenarios.

**Strengths:**

1. In response to the limitations of existing methods, BiAN is specifically designed for object counting tasks. The introduction of conditional alignment and conditional consistency mechanisms effectively addresses task-related factors such as object density changes, leading to improved counting accuracy.
2. The authors offer a detailed theoretical analysis, proving that BiAN achieves a lower bound on joint decision error. This analysis illustrates how conditional alignment contributes to error reduction, thereby providing strong theoretical support for the method’s effectiveness.
3. The authors achieved significant performance improvements.

**Weaknesses:**

1. On the object counting task, BiAN clearly achieves significant performance compared to other methods using DA. As the authors state, they only align feature distributions of objects of interest so that inter-subject information can be preserved. The authors need to provide more ablation experiments and analysis to ensure that the improvement in experimental results comes from retaining the target feature distribution.
2. In Tables 6-8, the author provides detailed structures and parameters. The author should consider adding network visualization.
3. The method proposed by the author is universal for the problem of domain adaptation learning. Does the author consider multi-task verification on other tasks?
4. This paper needs to be improved. For example, many quotation signs are wrong in sec. 4.1.

**Questions:**

see weaknesses

---

### Public Comment · ~Calvin_McCarter1 · 2024-11-16
**Relation to Confounded Domain Adaptation**

How does this work relate to the assumptions and framework described in [1]? There are, of course, major differences: it focuses on adapting features for downstream analytics tasks, while your work focuses on aligning representations for downstream object counting. But it does seem related, in that both are in some way minimizing a weighted average of some divergence between conditional distributions. See especially Section 4.5, which also uses image segmentation, then adapts while conditioning on segments, in that case to improve image color adaptation.

[1] Towards Backwards-Compatible Data with Confounded Domain Adaptation (TMLR 2024) https://openreview.net/pdf?id=GSp2WC7q0r

---

### Meta-Review · Area_Chair_rqXm · 2024-12-15

**Metareview:**

This paper receives 4 negative ratings and 1 positive rating. Although the paper has some merits, e.g., competitive results with theoretical analysis, the reviewers pointed out a few critical concerns about 1) technical clarity, 2) technical novelty compared to the prior domain alignment work, 3) more ablation experiments and analysis. After taking a close look at the paper, rebuttal, and discussions, the AC agrees with reviewers' feedback and hence suggests the rejection decision. The authors are encouraged to improve the paper based on the feedback for the next venue.

**Additional Comments On Reviewer Discussion:**

In the rebuttal, some of the concerns like technical clarity are explained by the authors. However, during the post-rebuttal discussion period, the reviewer rWXH is still concerned about the technical setting, while the reviewer KYiB and jo5q are not convinced by the experimental results and technical contributions explained by the authors. The AC agrees with the reviewers that these issues should be improved in the manuscript, which still requires a significant amount of effort.

---

### Decision · Program_Chairs · 2025-01-22

Reject